# Toward Preference-aligned Large Language Models via Residual-based Model Steering

## Abstract

Preference alignment is a critical step in making Large Language Models (LLMs) useful and aligned with (human) preferences. Existing approaches such as Reinforcement Learning from Human Feedback or Direct Preference Optimization typically require curated data and expensive optimization over billions of parameters, and eventually lead to persistent task-specific models. In this work, we introduce Preference alignment of Large Language Models via Residual Steering (PALRS), a training-free method that exploits preference signals encoded in the residual streams of LLMs. From as few as one hundred preference pairs, PALRS extracts lightweight, plug-and-play steering vectors that can be applied at inference time to push models toward preferred behaviors. We evaluate PALRS on various small-to-medium-scale open-source LLMs, showing that PALRS-aligned models achieve consistent gains on mathematical reasoning and code generation benchmarks while preserving baseline general-purpose performance. Moreover, when compared to DPO-aligned models, they perform better with huge time savings. Our findings highlight that PALRS offers an effective, much more efficient and flexible alternative to standard preference optimization pipelines, offering a training-free, plug-and-play mechanism for alignment with minimal data.

## 1 Introduction

Large Language Models (LLMs) have rapidly advanced the state-of-the-art performance across various domains, including dialogue, programming, and mathematical tasks (Li et al., 2025). While most capabilities in such systems are due to rich and wide pretraining (Chen et al., 2024; Kirchenbauer et al., 2024; Shaib et al., 2024; Wang et al., 2025b), a key determinant in their usability is how close their outputs align with *human preferences* (Wang et al., 2023; Shen et al., 2023). Indeed, preference alignment has emerged in recent years as a focal stage in the LLM deployment pipeline: approaches such as reinforcement learning from human feedback (Ouyang et al., 2022; Bai et al., 2022) and direct preference optimization (DPO) (Rafailov et al., 2023) have become standard practices for eliciting better capabilities from LLMs.

Despite their tangible effects, preference-optimization alignment methods remain costly and inflexible. First, current approaches rely on large volumes of curated preference datasets (Köpf et al., 2023), thus being highly *annotation intensive*. Second, despite parameter-efficient methods such as LoRA adapters, aligning a model remains *computationally intensive* because it requires repeated forward and backward passes through large models, often consuming substantial GPU hours. (Stiennon et al., 2020). Third, alignment is typically considered *persistent*: once an LLM has been fine-tuned toward a particular preference setting, adapting it to a different set of preferences generally requires starting again from the base model to produce new checkpoints. Maintaining multiple preference-specific checkpoints can quickly become resource-intensive. These challenges underscore the need to scale alignment methods across three dimensions: efficiency (reducing computational cost), effectiveness (maximizing alignment quality), and flexibility (enabling rapid adaptation to new preference specifications).

Recently, an emerging line of research has unveiled that *residual stream activations* of LLMs encode contextually rich and linearizable features that can be used to manipulate the model behavior surgically, yet without altering their weights or requiring any additional training (Zou et al., 2023; Zhang & Nanda, 2024). Residual-based interventions have been shown effective to mitigate refusal behav-

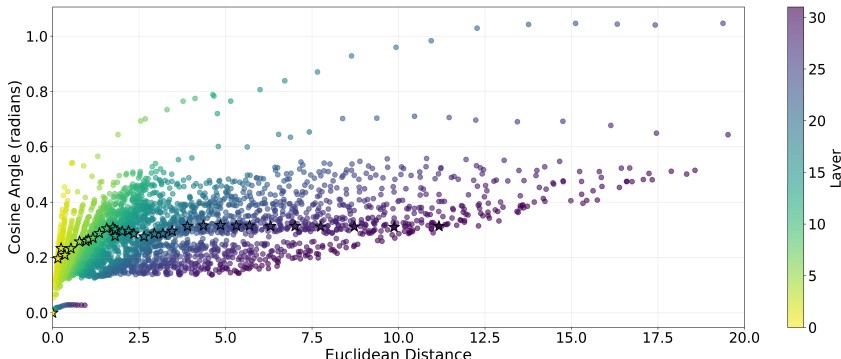

Figure 1: Euclidean distance and cosine angle between residual stream activations (from Llama 3.1 8B Instruct) of pairs of chosen-rejected responses to math questions, for a fixed token position (last position in each input response) and varying model layers. Stars denote average coordinates for each layer. *(Best viewed in color)*

iors (Arditi et al., 2024; Wang et al., 2025a), erase concepts (Belrose et al., 2023), induce desired persona-like behaviors (Chen et al., 2025), or improve factfulness (Li et al., 2023). These results suggest a promising direction, with residuals functioning as "control knobs," providing a relatively inexpensive yet effective mechanism for steering model behavior at inference time. However, prior work has focused on optimizing narrow behaviors (e.g., refusal mitigation), leaving the broader challenge of using residual interventions to align models with a range of preferences underexplored.

**Our Hypothesis.** Consider the task of improving a model's mathematical reasoning or coding capabilities. Standard practices based on preference optimization alignment, would require curating thousands of preference pairs, optimizing the model's weights, and deploying task-specific checkpoints, resulting in a resource-intensive and inflexible process.

In this work, we hypothesize that the difference between the residual stream activations of chosen and rejected responses—to questions grounded in a particular domain, e.g., math or coding—can be distilled into steering directions that can be used to induce *preferred behaviors* in a model via lightweight inference-time interventions.

To support our hypothesis, consider Figure 1, which shows Euclidean distance and cosine angle between residual stream activations (from Llama 3.1 8B Instruct) of pairs of chosen-rejected responses to math questions. The observed relatively large Euclidean distances offer evidence that the chosen and rejected activations can be far apart in residual space, which implies the possibility of defining a vector, or steering direction, that moves the activation from the rejected toward the chosen response: if the difference were tiny, a steering vector would have little effect, by contrast, large distances suggest the difference is sufficiently large to guide model behavior adjustment. In addition, the fairly low angles observed in the majority of points offer evidence that the differences between chosen and rejected activations are mostly consistent in direction, thus allowing a generalizable aggregated steering vector to be distilled, rather than needing a separate vector for each example. Combining the two insights, i.e., the chosen-rejected differences are significant in magnitude and relatively coherent in direction, raises a pattern suggesting that a residual-based intervention could reliably guide the model toward preferred behaviors.

**Preference alignment of Large Language Models via Residual Steering (PALRS).** Building on the above remarks, we introduce a novel approach to preference alignment of LLMs through steering with residual stream activations, dubbed PALRS. Instead of updating model weights via gradient optimization, PALRS identifies preference directions by extracting differences in residual activations from only a small set of preference pairs (on the order of 100). These directions are then applied at inference time, enabling lightweight, plug-and-play steering toward preferred behaviors.

To the best of our knowledge, this is the first study to leverage residual stream directions for preference alignment. Our contributions are threefold:

- We bring the model steering framework based on residual stream activations to the setting of preference alignment of LLMs, showing that residual stream activations encode preference information in a linearly accessible form.

- We introduce a simple difference-in-means approach for estimating candidate preference directions from a small set of chosen–rejected response pairs, without requiring any LLM post-training. We further propose a principled criterion for selecting the steering direction that best aligns an LLM's behavior with target preferences.

- We demonstrate the effectiveness of PALRS through different small-to-medium scale open-source LLMS, with testing on widely used benchmarks. Particularly, as a concrete case study, we show that steering directions derived from preference-alignment datasets conceived for *math reasoning* (GSM8K) and *code generation* (HumanEval), under certain conditions of the steering intensity, lead an LLM to improve performance on corresponding benchmarks, without degrading results on other-domain benchmarks. Additionally, we highlight that PALRS-aligned models outperform DPO-aligned models on both GSM8K and HumanEval, achieving superior effectiveness while requiring much less computational overhead.

## 2 METHODOLOGY

### 2.1 PRELIMINARIES

**Notation.** Throughout this paper, we will use capital letters to denote data objects, lowercase letters to denote scalars, and bold lowercase letters to denote vectors.

We are given a collection of text triplets $\langle Q, A^{(+)}, A^{(-)} \rangle$ where $Q$ is a question, and $A$s are two possible (human-provided) answers to $Q$. Based upon this collection, we define the dataset $\mathcal{D} = \{\langle Q, A^{(+)}, A^{(-)} \rangle \mid A^{(+)} \succ_Q A^{(-)}\}$, where $A^{(+)} \succ_Q A^{(-)}$ denotes that, for question $Q$, $A^{(+)}$ is preferred over $A^{(-)}$. Let also $\mathbf{t}^{(+)}$, $\mathbf{t}^{(-)}$, and $\mathbf{t}^{(Q)}$ denote the input sequences of tokens from an answer $A^{(+)}$, $A^{(-)}$[1] and question $Q$, respectively.

**Residual stream activations.** Following previous studies (Zhang & Nanda, 2024), we resort to the concept of *residual stream activation*, specifically in the context of a standard Transformer decoder model, with $L$ layers and hidden size $d$.

Given an input sequence $\mathbf{t}$, the state of knowledge a model has about a token in position $i$ at the start of layer $\ell$ (i.e., before layer $\ell$ processes it) can effectively be expressed by the token's residual stream activation, given the input tokens up to position $i$ and all contributions computed from the layers preceding $\ell$. We will denote it as a real-valued $d$-dimensional vector $\mathbf{x}_{i,\ell}(\mathbf{t})$, or simply with $\mathbf{x}_{i,\ell}$ if $\mathbf{t}$ is clear from the context.

In other words, the token's residual stream activation is the accumulated hidden representation of the token as it flows through the model, thus encoding all contextual information that the model has built up so far—which includes the initial token and positional embeddings, plus the contributions from the multi-head causal self-attention and feed-forward MLP components (sublayers) of all previous layers. This also means that the residual stream is linearly interpretable, since sublayer outputs are added to the residual stream, and is tied to the model's prediction distribution, since at the final layer the model projects the last residual stream through the embedding-to-token matrix to get the next-token logits.

**Model steering.** The residual stream activation can be used for model steering because it is the central state vector that encodes all of the model's knowledge about a token at a given point.

One effective way to reliably alter the model's outputs and behavior is *activation addition*, i.e., adding an identified direction $\mathbf{r} \in \mathbb{R}^d$ in the residual space that corresponds to some feature, e.g., tokens that are more representative of the chosen responses than the rejected ones. Therefore, shifting the residual stream along that direction will change the model's predictions accordingly.

---

[1]The two vectors may have arbitrary and different lengths; however, it will be ensured that any token positions remain consistent across both vectors.

An opposite approach is *directional ablation*, which consists in subtracting from the residual stream its orthogonal projection onto the direction $\mathbf{r}$. However, as noted in (Arditi et al., 2024), the two approaches have a different impact as the directional ablation applies to all layers and token positions; by contrast, the activation addition involves only a desired layer (and applies across all token positions). Besides that, while ablation has been proven effective in refusal-removal setups (Arditi et al., 2024; Wang et al., 2025a), it is not well-suited to our setting: indeed, chosen and rejected responses typically differ only by a few key tokens, thus ablating their direction is very likely to disrupt the model's behavior.

## 2.2 Extracting Preference Directions

To extract the candidate *preference directions* from the model's residual stream activations, we resort to the *difference-in-means* approach (Belrose, 2023), which has shown to be effective in previous work (Arditi et al., 2024; Tigges et al., 2024; Wang et al., 2025a).

Given the dataset $\mathcal{D} = \{\langle Q, A^{(+)}, A^{(-)} \rangle \mid A^{(+)} \succ_Q A^{(-)}\}$ of triplets of questions and chosen-rejected responses, we first compute two quantities for any choice of token position $i$ and layer $\ell$, which correspond to the average of the residual stream activations produced by the model when it receives in input the chosen, resp. rejected, responses:

$$\boldsymbol{\mu}_{i,\ell}^{(+)} = \frac{1}{|\mathcal{D}|} \sum_{A^{(+)} \in \mathcal{D}} \mathbf{x}_{i,\ell}(\mathbf{t}^{(+)}), \qquad \boldsymbol{\mu}_{i,\ell}^{(-)} = \frac{1}{|\mathcal{D}|} \sum_{A^{(-)} \in \mathcal{D}} \mathbf{x}_{i,\ell}(\mathbf{t}^{(-)}). \tag{1}$$

We define the *candidate preference direction* for a given token position $i$ and layer $\ell$ as the difference between the two means as follows:

$$\mathbf{r}_{i,\ell} = \boldsymbol{\mu}_{i,\ell}^{(+)} - \boldsymbol{\mu}_{i,\ell}^{(-)}. \tag{2}$$

It is worth noticing that, to prevent diluting the residual stream signals, we focus on the chosen and rejected responses while discarding the instruction $Q$ from the computation of directions. Indeed, we are interested in discerning the signals that differentiate the chosen tokens from the rejected ones, regardless of a particular prompt.

**Position and layer selection strategies.** Following (Arditi et al., 2024), we consider only token positions corresponding to *post-instruction tokens*, i.e., the template tokens following the instruction (e.g., `<|eot_id|>` in Llama 3, cf. Appendix A), ensuring the model processed the given text and starts producing its output.

Regarding the selection of layers, we emphasize the importance of focusing on mid-to-late layers for model steering. The rationale is that early layers primarily shape broad syntactic and structural features, well before semantics, knowledge retrieval, and reasoning processes emerge in the mid layers; also, late layers and especially the unembedding layer directly influence the logits, which bias outputs without meaningfully altering intermediate reasoning.

## 2.3 Selecting the Steering Direction

Our goal is to choose the vector that can effectively steer the model toward the desired preference-aligned behavior, i.e., favoring human-based preferences in generating responses, while avoiding disruption of its general capabilities.

First, we consider the average signal of residual stream activations induced by the chosen responses from $\mathcal{D}$. Specifically, for a given layer $\ell$, we compute the mean residual stream activation at each selected token position and then take the average across all such positions. We denote this quantity by $\boldsymbol{\mu}_{\ell}^{(+)}$.

Let us denote with $\mathcal{C} = \{(i, \ell)\}$ the set of pairs (token-position, layer id) that are selected as result of the previous stage of position and layer selection. We aim to select the steering direction by finding the preference vector $\mathbf{r}_{i,\ell}$ (with $(i, \ell) \in \mathcal{C}$) that is most strictly aligned with $\boldsymbol{\mu}_{\ell}^{(+)}$, that is the vector $\mathbf{r}_{i,\ell}$ that preserves the direction of $\boldsymbol{\mu}_{\ell}^{(+)}$ exactly, while only boosting its magnitude. This can be

accomplished by finding the vector projection of a $\mathbf{r}_{i,\ell}$ onto $\boldsymbol{\mu}_\ell^{(+)}$ with the maximum magnitude, or simply the one maximizing the scalar product with $\boldsymbol{\mu}_\ell^{(+)}$:

$$\mathbf{r}^* = \arg \max_{(i,\ell) \in \mathcal{C}} \left\| \left( \mathbf{r}_{i,\ell} \cdot \frac{\boldsymbol{\mu}_\ell^{(+)}}{\|\boldsymbol{\mu}_\ell^{(+)}\|} \right) \frac{\boldsymbol{\mu}_\ell^{(+)}}{\|\boldsymbol{\mu}_\ell^{(+)}\|} \right\| = \arg \max_{(i,\ell) \in \mathcal{C}} \left| \mathbf{r}_{i,\ell} \cdot \boldsymbol{\mu}_\ell^{(+)} \right|. \tag{3}$$

Above, the absolute value ensures that only the magnitude of alignment with $\boldsymbol{\mu}_\ell^{(+)}$ is maximized, thereby avoiding negative contributions that would otherwise diminish $\|\boldsymbol{\mu}_\ell^{(+)}\|$.

Once we have best aligned a preference direction with the one of the mean activations of the chosen responses, we rescale $\mathbf{r}^*$ so that its norm matches $\|\boldsymbol{\mu}_{\ell^*}^{(+)}\|$, where $\ell^*$ denotes the layer corresponding to one of the selected $\mathbf{r}^*$:

$$\hat{\mathbf{r}}^* = \frac{\|\boldsymbol{\mu}_{\ell^*}^{(+)}\|}{\|\mathbf{r}^*\|} \cdot \mathbf{r}^*. \tag{4}$$

The above transformation makes the two vectors comparable on the same scale, which allows us to better control the effect of the multiplicative factor in the activation addition step, as described next.

## 2.4 APPLYING THE STEERING DIRECTION

The selected prefernce direction $\mathbf{r}^*$ is eventually used to steer the model toward preference-aligned behavior. As discussed in Sect. 2.1, the model steering is performed through activation addition, which means that the selected preference direction is added to the residual stream activations of any newly generated response by the model. Specifically, given an input token sequence $\mathbf{t}$ and by denoting with $\mathbf{x}_{\ell^*}$ the residual stream activations at any token position and at layer $\ell^*$, the steered residuals are defined as follows:

$$\mathbf{x}'_{\ell^*} := \mathbf{x}_{\ell^*}(\mathbf{t}) + \alpha \hat{\mathbf{r}}^*, \tag{5}$$

where $\alpha \in \mathbb{R}^+$ is a coefficient that controls the strength of the steering effect.

## 2.5 EXPERIMENTAL SETUP

**Preference datasets.** To demonstrate our proposed PALRS approach, we chose to focus on two particularly informative testbeds for alignment, namely mathematical reasoning and coding. In fact, compared to broad, general-purpose tasks like commonsense reasoning, math reasoning and coding demand precise logical reasoning and adherence to rules—small alignment errors can directly break correctness—and models' responses can usually be evaluated unambiguously as correct or incorrect.

Within this view, we used the *argilla/distilabel-math-preference-dpo* dataset,[2] which provides chosen/rejected pairs grounded in mathematical correctness, and *inclusionAI/Ling-Coder-DPO*,[3] whose preferences aim at improving correctness in coding generation.

From each dataset, we randomly sampled 100 triplets to construct the collection $\mathcal{D}$, which we use to compute residual stream activations of chosen and rejected responses. Note that the decision to sample a relatively small number of instances is deliberate, as this is sufficient to capture the difference-in-means signal, in line with findings from related work (Arditi et al., 2024; Wang et al., 2025a). In addition, unlike prior work using residual vectors for refusal ablation (Arditi et al., 2024; Wang et al., 2025a) or personality steering (Chen et al., 2025), our approach does not rely on external cues for sample selection, such as evaluation scores, targeted refusal tokens, or LLMs-as-judges.

**Evaluation goals and benchmarks.** We define the following evaluation goals:
- **(E1)** – Assessing the performance of PALRS-aligned models against baseline models (i.e., not steered) using two well-established benchmarks for the target tasks: GSM8K (Cobbe et al., 2021) for mathematical reasoning and HumanEval (Chen et al., 2021) for code generation.
- **(E2)** – Assessing the performance of PALRS-aligned models on tasks outside the target domains,

---

[2] https://huggingface.co/datasets/argilla/distilabel-math-preference-dpo
[3] https://huggingface.co/datasets/inclusionAI/Ling-Coder-DPO

| Model | Hugging Face ID | Params | Post-training Strategy | No. Layers |
|---|---|---|---|---|
| Llama 1B | meta-llama/Llama-3.2-1B-Instruct | 1B | SFT + RLHF | 16 |
| Llama 3B | meta-llama/Llama-3.2-3B-Instruct | 3B | SFT + RLHF | 28 |
| Llama 8B | meta-llama/Llama-3.1-8B-Instruct | 8B | SFT + RLHF | 32 |
| Mistral | mistralai/Mistral-7B-Instruct-v0.3 | 7B | SFT | 32 |
| OLMo | allenai/OLMo-2-1124-7B-Instruct | 7B | SFT + DPO + RLVR | 32 |

Table 1: Summary of used LLMs. Post-training strategies are abbreviated as SFT: Supervised Fine-Tuning, RLHF: Reinforcement Learning From Human Feedback, RLVR: Reinforcement Learning with Verifiable Reward, DPO: Direct Preference Optimization.

using five widely adopted benchmarks: ARC-Challenge (Bhakthavatsalam et al., 2021) for scientific and commonsense reasoning, HellaSwag (Zellers et al., 2019) for physical and social commonsense inference, MMLU (Hendrycks et al., 2021) for broad knowledge and multi-task understanding, TruthfulQA (Lin et al., 2022) for factuality and truthfulness, and WinoGrande (Sakaguchi et al., 2020) for coreference resolution and pronoun disambiguation. By treating these benchmarks as *guardrails*, this evaluation aims to measure whether, and to what extent, PALRS-aligned models for math or coding steering are able to preserve general-purpose capabilities of baseline models.

• **(E3)** – Comparing PALRS-aligned models with DPO-aligned models on the target tasks, using the same 100-sample preference data, and evaluating their performance on the benchmarks as well as their efficiency.

• **(E4)** – Assessing the parameter sensitivity of the steering coefficient and its impact on PALRS in the target tasks.

To ensure reliability and reproducibility in the evaluation of models, we used the well-established Language Model Evaluation Harness framework (Gao et al., 2024) via its TinyBenchmarks tasks (Polo et al., 2024), which are a curated selection of samples from the aforementioned benchmarks that ensure the same evaluation robustness as the full one, at a reduced temporal cost. The performance results which will be reported for the experiments correspond to *exact_match* for GSM8K, *pass@1* for HumanEval, and *accuracy* for the remaining benchmarks.

**Models.** We experimented with a range of LLMs differing in family, post-training strategy, and parameter size, as summarized in Table 1. These include Llama3 (Dubey et al., 2024) in its 1B, 3B, and 8B variants, Mistral (Jiang et al., 2023) 7B, and OLMo2 (OLMo et al., 2025) 7B. This variety enables us to assess to some extent the impact of model architectures, post-training approaches, and sizes on the steering behavior induced by PALRS.

As mentioned in Sect. 2.2, we inspected mid-to-late layers in each of the models. Specifically, we selected a range $[0.3L..0.9L]$, where $L$ is the number of layers as reported in Table 1.

## 3 RESULTS

**Performance of PALRS on target tasks (E1).** Looking at the first two result-columns in Table 2,[4] we observe how $\text{PALRS}_{Math}$ systematically boosts the math-related task (GSM8K) across all models, with an average improvement over the baseline around +14% (from +3.7% with OLMo up to +20.1% with Llama 1B). Analogously, $\text{PALRS}_{Code}$ consistently improves the code-related task (HumanEval) across all models, with an average improvement over the baseline around +22% (from +2.6% with Llama 8B up to +53.3% with Mistral).

**Performance of PALRS on guardrail tasks (E2).** Important insights also emerge from the performance over the guardrail benchmarks (right-most five columns in Table 2). PALRS-aligned Llama 1B, Llama 8B, and OLMo are fairly stable, with average percentage changes on guardrail benchmarks around -1% or less—specifically, for $\text{PALRS}_{Math}$ resp. $\text{PALRS}_{Code}$, -0.86% resp. +0.35% with Llama 1B, -0.64 resp. -0.08% with Llama 8B, and -0.73% resp. -0.51% with OLMo. $\text{PALRS}_{Math}$ with Mistral even slightly improves on average over the guardrails (+0.26%), while

---

[4]Table 2 reports the results corresponding to the best model configurations for steering direction extraction, as reported in Table 4 in Appendix C.

| Tasks | | Targets ↑ | | Guardrails ↑ | | | | |
|---|---|---|---|---|---|---|---|---|
| **Model** | **Variant** | GSM8K | HumanEval | ARC-C | HellaSwag | MMLU | TruthQA | WinoGrande |
| Llama 1B | Baseline | 0.34 | 0.38 | 0.44 | 0.55 | 0.43 | 0.43 | 0.57 |
| | PALRS$_{Math}$ | **0.41** | 0.41 | 0.42 | 0.54 | 0.44 | 0.42 | 0.58 |
| | | (+20.10%) | (+7.89%) | (-3.68%) | (-1.03%) | (+2.39%) | (-2.89%) | (+0.90%) |
| | PALRS$_{Code}$ | 0.31 | **0.48** | 0.41 | 0.56 | 0.45 | 0.44 | 0.56 |
| | | (-9.90%) | (+26.32%) | (-6.70%) | (+1.95%) | (+4.32%) | (+3.64%) | (-1.47%) |
| Llama 3B | Baseline | 0.62 | 0.60 | 0.57 | 0.78 | 0.63 | 0.48 | 0.61 |
| | PALRS$_{Math}$ | **0.70** | 0.56 | 0.54 | 0.73 | 0.62 | 0.45 | 0.62 |
| | | (+13.53%) | (-6.67%) | (-5.54%) | (-5.60%) | (-0.85%) | (-5.61%) | (+1.61%) |
| | PALRS$_{Code}$ | 0.57 | **0.67** | 0.55 | 0.74 | 0.61 | 0.48 | 0.64 |
| | | (-7.72%) | (+11.67%) | (-3.66%) | (-5.19%) | (-2.89%) | (-0.39%) | (+4.52%) |
| Llama 8B | Baseline | 0.72 | 0.78 | 0.65 | 0.81 | 0.63 | 0.54 | 0.75 |
| | PALRS$_{Math}$ | **0.82** | 0.77 | 0.63 | 0.81 | 0.63 | 0.55 | 0.73 |
| | | (+13.09%) | (-1.28%) | (-3.32%) | (+0.32%) | (-0.12%) | (+1.62%) | (-1.71%) |
| | PALRS$_{Code}$ | 0.76 | **0.80** | 0.65 | 0.81 | 0.63 | 0.54 | 0.75 |
| | | (+4.71%) | (+2.56%) | (-0.52%) | (+0.00%) | (-0.58%) | (+0.18%) | (+0.50%) |
| Mistral | Baseline | 0.45 | 0.15 | 0.64 | 0.84 | 0.64 | 0.61 | 0.76 |
| | PALRS$_{Math}$ | **0.53** | 0.15 | 0.65 | 0.84 | 0.64 | 0.61 | 0.76 |
| | | (+18.42%) | (+0.00%) | (+1.64%) | (-0.05%) | (-0.19%) | (-0.09%) | (+0.00%) |
| | PALRS$_{Code}$ | 0.47 | **0.23** | 0.65 | 0.84 | 0.64 | 0.61 | 0.75 |
| | | (+3.84%) | (+53.33%) | (+0.95%) | (+0.02%) | (-0.10%) | (-1.46%) | (-1.31%) |
| OLMo | Baseline | 0.75 | 0.52 | 0.66 | 0.83 | 0.62 | 0.56 | 0.76 |
| | PALRS$_{Math}$ | **0.77** | 0.53 | 0.65 | 0.83 | 0.61 | 0.54 | 0.77 |
| | | (+3.71%) | (+1.92%) | (-1.19%) | (-0.18%) | (-1.51%) | (-2.43%) | (+0.72%) |
| | PALRS$_{Code}$ | 0.78 | **0.60** | 0.66 | 0.83 | 0.62 | 0.55 | 0.75 |
| | | (+4.88%) | (+15.38%) | (+0.26%) | (+0.17%) | (+0.00%) | (-1.71%) | (-1.28%) |

Table 2: Benchmark performance results: Baseline vs. PALRS-aligned models. Bold values correspond to PALRS-aligned model performances on the target tasks. Values in parenthesis indicate percentage increase of a PALRS-aligned model w.r.t. the baseline performance on the same task.

keeping average guardrail degradation around -0.38% when using PALRS$_{Code}$. By contrast, Llama 3B is the only model to suffer the most guardrail degradation, up to -3.19% with PALRS$_{Math}$.

Focusing on the models, Mistral provides the largest boost by PALRS$_{Code}$ (+53.4%) and second large by PALRS$_{Math}$ (+18.4%). Using Llama models has shown a scaling effect, since smaller models lead to relative larger gains but also stronger impact on guardrail tasks. By contrast, PALRS-aligned OLMo models have the least improvement (+3.7%) over the baseline on the math task, but a significant +15.4% on the coding task.

Remarkably, PALRS alignment of the largest models, i.e., OLMo, Mistral and Llama 8B, on one target task reveals little degradation or, more often, improvement over the other target task, suggesting that steering in particular on a math task can have beneficial effect on a coding task as well.

**Comparison of PALRS with DPO (E3).** Figure 2 compares PALRS-aligned models with DPO-aligned models, in terms of effectiveness and efficiency on the GSM8K and HumanEval benchmarks (cf. Appendix B for details on the hardware used). PALRS$_{Math}$-aligned models always outperform the corresponding DPO-aligned on GSM8K: we notice a clear percentage increase consistently over all models, ranging from +2.6% with OLMo to +20.5% with Mistral, with an average gain of +10.4%, thus showing robustness across model-scales.

This couples with another outstanding result, which regards time efficiency: PALRS$_{Math}$-aligned models are an order of magnitude faster than the corresponding DPO-aligned on GSM8K. For example, on Llama 3B, learning the PALRS$_{Math}$-aligned model takes about 31s vs. DPO-aligned one's 300s, i.e., $\sim 10x$ faster. It is also worth emphasizing that, as models grow larger, DPO-aligned

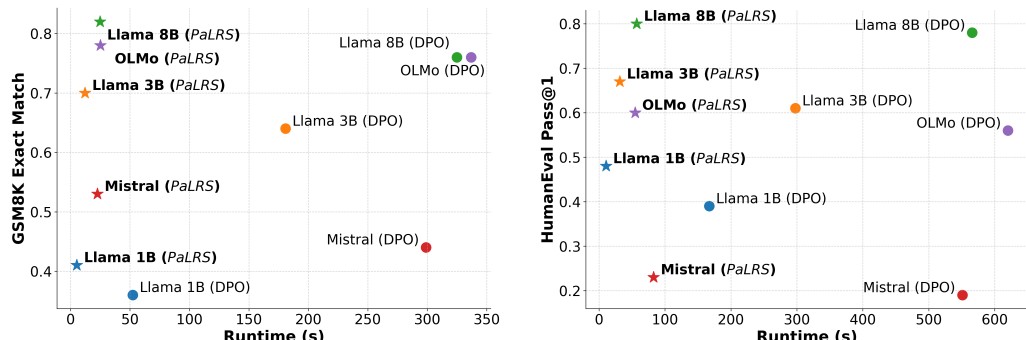

Figure 2: Benchmark performance and efficiency comparison of PALRS- and DPO-aligned models: GSM8K (left) and HumanEval (right). Colored points denote models, star resp. circle markers denote PALRS- resp DPO-alignment.

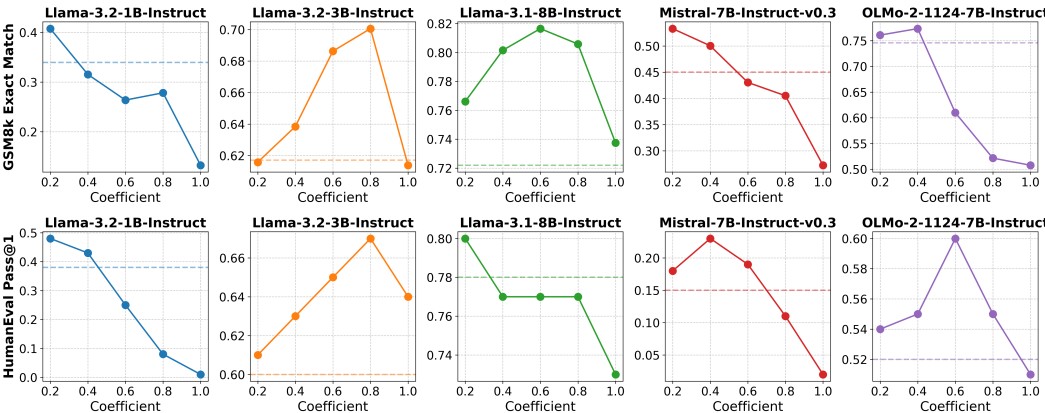

Figure 3: Performance of PALRS-aligned models on the target tasks GSM8K (top) and HumanEval (bottom) by varying the steering coefficient. Dashed horizontal line marks the baseline performance.

models' times increase steeply, while PALRS$_{Math}$-aligned models scale much more smoothly (from 5s to about 25s). Accuracy also scales better with PALRS$_{Math}$-aligned models: the performance gap against DPO widens with larger models.

Analogous remarks can be drawn from the comparison on the HumanEval benchmark. The performance gain of PALRS vs. DPO here is from +2.6% (Llama 8B) to 23.1% (Llama 1B), and tends to diminish as models get larger, again except for Mistral, whereby PALRS$_{Code}$ model has a percentage increase of +21%. Also, PALRS$_{Code}$-aligned models are again an order of magnitude faster than corresponding DPO models, with an average speedup of 10x across models, and the time gap increases linearly with model size, showing PALRS$_{Code}$-aligned models are far more scalable.

**Impact of the steering coefficient (E4).** We investigate the sensitivity of the steering coefficient $\alpha$ (cf. Eq. (5)) and its effect on the performance of PALRS-aligned models on the target tasks. Figure 3 provides insights which can be summarized as follows. We notice that the coefficient sensitivity is model- and task-dependent. While in general performance trends drop consistently as the coefficient increases, there is always a regime of the coefficient where the PALRS-aligned models outperform the relative baselines. Low to moderate coefficients (up to 0.8) often yield the best trade-offs, especially for Llama 3B, Llama 8B (GSM8K) and OLMo (HumanEval). Oversteering is most visible at coefficient at 1.0 for all models. More specifically, for Llama 3B and OLMo (HumanEval), the beneficial range is around 0.6–0.8 before oversteering occurs; for Mistral and Llama 1B, oversteering happens much earlier (as low as 0.4–0.6), while Llama 8B tolerates higher coefficients better but still show oversteering when pushed too far. To sum up, steering shows to be beneficial, but needs moderate tuning of the coefficient to avoid oversteering and performance degradation.

## 4 RELATED WORK

**Model Steering via Feature Directions.** Recent studies suggest that linear directions in the activation space of LLMs capture richer and more generalizable features than individual neurons (Bolukbasi et al., 2016; Li et al., 2021; Elhage et al., 2022). This shift toward subspace-level aspects has motivated research that exploits linear representations to probe model internals and provide better interpretation and steering of their behaviors (Hernandez & Andreas, 2021; Nanda et al., 2023; Park et al., 2024). A key challenge is to reliably identify such feature directions. In this regard, unsupervised approaches based on Sparse Auto-Encoders (SAE) have been used to uncover latent, interpretable features (Huben et al., 2024; Lan et al., 2024; Shu et al., 2025). A complementary trend involves exploiting contrastive pairs of texts differing across a specific axis (e.g., sentiment) to extract directions that isolate targeted behaviors (Burns et al., 2023; Rimsky et al., 2024).

Once extracted, these features can be used to intervene on model activations, particularly in the residual stream, where editing is known to effectively steer the models' behavior (Zou et al., 2023; Zhang & Nanda, 2024). Such interventions have been shown promising in shifting sentiment and detoxification (Turner et al., 2023), enhancing truthfulness (Li et al., 2023), erasing concepts (Belrose et al., 2023), targeting refusal behaviors (Arditi et al., 2024; Wang et al., 2025a), and controlling character traits in LLMs Chen et al. (2025).

**Preference Optimization in LLMs.** A large body of work has focused on aligning LLMs with human preferences, aiming to render such tools more usable Wang et al. (2023); Shen et al. (2023). This has been largely driven by Reinforcement Learning from Human Feedback (RLHF) approaches (Ouyang et al., 2022; Bai et al., 2022), and more efficient methods like Direct Preference Optimization (DPO) (Rafailov et al., 2023). Nonetheless, despite recent efforts to further improve efficiency (Hong et al., 2024; Meng et al., 2024), the promising and more efficient alternative of steering models toward preferences via residual streams remained almost totally unexplored.

In a related effort, a recent work by Liu et al. (2024) explores preference alignment by identifying disparities in activation patterns elicited by preferred versus dispreferred stimuli. While this represents an important step toward bridging preference data with internal model representations, their approach differs fundamentally from ours. First, it requires training with contrastive stimuli to extract the relevant signals, whereas our method is entirely *training-free*. Second, it introduces a low-rank adaptation module to perform steering, in contrast to our *inference-time intervention*. These design choices also manifest in practice, as their method yields more limited steering effects compared to the lightweight and scalable improvements enabled by PALRS.

## 5 CONCLUSIONS

In this work, we presented PALRS, a training-free method for preference alignment that leverages residual stream activations to steer LLM behavior directly at inference time. By distilling steering vectors from as few as a hundred preference pairs, PALRS aligns models with desired behaviors without any parameter updates, costly optimization, or the need to maintain multiple fine-tuned checkpoints. Crucially, our results show that PALRS-aligned models are a safer and consistently superior alternative to their DPO-aligned counterparts: never underperforming, often achieving substantial gains, and doing so with orders-of-magnitude less computation. Our findings render residual-based steering as a powerful paradigm for preference alignment, aiming to make it simpler, more effective yet scalable, and broadly accessible compared to traditional post-training approaches.

**Limitations.** While our results highlight the promise of residual-based preference alignment, several limitations remain. First, although we evaluated PALRS on a relatively representative sample of model families and post-training modalities, its generalizability to larger-scale or proprietary models needs to be evaluated. Second, our current method for discovering effective data subsets and steering coefficients relies on heuristic grid search: developing principled, theoretically grounded selection strategies remains a key direction for smoother real-world deployment of our results. Third, while our results demonstrate that residual interventions effectively steer model behavior and provide empirical support for our hypothesis, more in-depth explainability of how preference information is encoded and disentangled across layers is needed. Overall, we demonstrate the feasibility of this approach, while leaving refinement and addressing these limitations to future research.

## ETHICS STATEMENTS

Our work demonstrates that residual-based preference steering can effectively influence model behavior. While this highlights the potential for beneficial applications, we acknowledge that such methods might also be misused to steer models toward harmful or malicious behaviors. We strongly discourage any such misuse and do not assume responsibility for applications beyond the scope of this research. We encourage researchers and practitioners within this research area to adhere to established safety and ethical guidelines when applying our proposed technique or related ones, and to prioritize transparency, fairness, and safety of end users.

**Reproducibility Statement.** We are strongly committed to ensuring reproducibility of our results. All experimental settings, hyperparameters, and model configurations are provided in detail in the paper or in the code repository of PALRS, which is available at `https://anonymous.4open.science/r/PaLRS-ICLR2026/`.

**Use of GenAI.** We disclose that GenAI assisted exclusively for light text editing. All intellectual contributions, ideas, methodology, experiments, and analyses, are solely attributable to the authors.

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

## A    CHAT TEMPLATES

For each considered model in our study, we resorted to the original chat template as reported either in the corresponding paper or in the HuggingFace model's page, as shown in Table 3.

| Model Family | Template |
|---|---|
| LLama3 | `<|begin_of_text|><|start_header_id|>user<|end_header_id|>instruction<|eot_id|><|start_header_id|>assistant<|end_header_id|>` |
| Mistral | `[INST] instruction [/INST]` |
| OLMo | `<|endoftext|><|user|>instruction<|assistant|>` |

Table 3: Chat templates for the model families considered in this study. Blue tokens indicate *post-instruction tokens*, as discussed in Section 2.2.

## B    DETAILS ON THE RUNNING ENVIRONMENT

We performed our experiments using an 8x NVIDIA A30 GPU server with 24 GB of RAM each, 764 GB of system RAM, a Double Intel Xeon Gold 6248R with a total of 96 cores, and Ubuntu Linux 20.04.6 LTS as OS.

## C  DETAILS ON THE MODEL CONFIGURATIONS USED FOR THE MAIN RESULTS

Table 4 reports the best model configurations used to obtain $\hat{\mathbf{r}}^*$, and the corresponding steering coefficient $\alpha$, used for the results shown throughout the main paper.

| Model | Math | | | | Code | | | |
|---|---|---|---|---|---|---|---|---|
| | seed | $i$ | $\ell^*/L$ | $\alpha$ | seed | $i$ | $\ell^*/L$ | $\alpha$ |
| Llama 1B | 870 | -2 | 14/16 | 0.2 | 343 | -5 | 14/16 | 0.2 |
| Llama 3B | 921 | -5 | 23/28 | 0.8 | 689 | -1 | 24/28 | 0.8 |
| Llama 8B | 245 | -4 | 28/32 | 0.6 | 94 | -4 | 26/32 | 0.2 |
| Mistral | 790 | -2 | 24/32 | 0.2 | 237 | -1 | 28/32 | 0.4 |
| OLMo | 311 | -3 | 23/32 | 0.4 | 447 | -3 | 26/32 | 0.6 |

Table 4: (Grid-search based) Best seed, token position ($i$), layer ($\ell^*/L$) and steering coefficient ($\alpha$) used throughout our experimental evaluation of PALRS-aligned models.

## D  CONFIGURATION FOR DPO ALIGNMENT

Table 5 reports the settings we used to align our used models with DPO for a comparison with PALRS.

| Hyperparameter | Value |
|---|---|
| Batch size (per device) | 1 |
| Gradient accumulation steps | 8 |
| Gradient checkpointing | Enabled |
| Learning rate | $1e$-6 |
| Scheduler | Linear |
| Optimizer | Paged AdamW (32-bit) |
| Training epochs | 3 |
| $\beta$ coefficient | 0.1 (default) |

Table 5: Training hyperparameters used for DPO.

## E  QUALITATIVE EXAMPLES OF PALRS'S STEERING EFFECT

Figures 4-7 provide qualitative insights into the effect of model steering toward preferred behaviors obtained via PALRS.

**Question:** *James has 30 teeth. His dentist drills 4 of them and caps 7 more teeth than he drills. What percentage of James' teeth does the dentist fix?*

**Expected Answer:**
First find the total number of teeth the dentist caps: 4 teeth + 7 teeth = **11 teeth**
Then find the total number of teeth the dentist fixes: 11 teeth + 4 teeth = **15 teeth**
Then divide that number by the total number of teeth and multiply by 100% to find the percentage of teeth he fixes: 15 teeth / 30 teeth * 100% = **50%**

**Baseline:** James has 30 teeth. The dentist drills 4 teeth. **So, the number of teeth that the dentist fixes is 30 - 4 = 26.**
The dentist caps 7 more teeth than he drills. So, the number of teeth that the dentist caps is 4 + 7 = 11.
The number of teeth that the dentist fixes is **26**. The number of teeth that the dentist caps is 11. So, the percentage of teeth that the dentist fixes is **26 / 30 * 100 = 86.67%**

**PALRS $_{Math}$:** James has 30 teeth. His dentist drills 4 teeth, so he has 30 - 4 = 26 teeth left.
The dentist caps 7 more teeth than he drills, so he caps 4 + 7 = **11 teeth**.
So James has 26 - 11 = **15 teeth** that the dentist fixes.
The percentage of teeth that the dentist fixes is **15/30 * 100% = 50%**

Figure 4: Example of mathematical improvement via PALRS for Llama 3.2 1B.

**Question:** *Mia and Emma are currently 16 years apart in age. If Mia, who is younger than Emma, is 40 years old, what's the average of their ages?*

**Expected Answer:**
If Mia is 40 years old, Emma is $40 + 16 = $ **56** years old.
The sum of their ages is $56 + 40 = $ **96** years.
The average age of the two is $96/2 = $ **48** years.

**Baseline:** Mia is 40 years old and Emma is **16 years old**, so the average age is $(40 + $ **16**$)/2 = $ **56 / 2 = 28** years.

**PALRS $_{Math}$:** Mia is 40 years old and Emma is $40 + 16 = $ **56** years old.
The average age of the two of them is $(40 + $ **56**$)/2 = $ **96 / 2 = 48** years.

Figure 5: Example of mathematical improvement via PALRS for Llama 3.2 1B.

**Question:**

```
def count_distinct_characters(string: str) -> int:
    """ Given a string, find out how many distinct
    characters (regardless of case) does it consist of
    >>> count_distinct_characters('xyzXYZ')
    3
    >>> count_distinct_characters('Jerry')
    4
    """
```

**Baseline:**

```
def count_distinct_characters(string: str) -> int:
    # Convert the string to lowercase and remove any whitespace
    string = string.lower().replace(" ", "")

    # Create a set from the string and return the length
    # of the set
    return len(set(string))
```

**PALRS** $_{Code}$**:**

```
def count_distinct_characters(string: str) -> int:
    # Create a set to store the characters
    char_set = set()

    # Iterate through the string and add each character
    # to the set
    for char in string:
        char_set.add(char.lower())

    # Return the size of the set
    return len(char_set)
```

Figure 6: Example of mathematical improvement via PALRS for Mistral 7B.

**Question:**

```python
def string_xor(a: str, b: str) -> str:
    """ Input are two strings a and b consisting
    only of 1s and 0s.
    Perform binary XOR on these inputs and return
    result also as a string.
    >>> string_xor('010', '110')
    '100'
    """
```

**Baseline:**

```python
def string_xor(a: str, b: str) -> str:
    result = ""
    for i in range(len(a)):
        if a[i] == b[i]:
            result += "0"
        else:
            result += "1"
    return result
```

**PALRS** $_{Code}$**:**

```python
def string_xor(a: str, b: str) -> str:
    # Initialize result string
    result = ""

    # Iterate through both strings
    for i in range(len(a)):
        # Perform XOR operation
        result += str(int(a[i]) ^ int(b[i]))

    return result
```

Figure 7: Example of mathematical improvement via PALRS for Mistral 7B.

