# OpenReview forum: "Toward Preference-aligned Large Language Models via Residual-based Model Steering"
_ICLR.cc/2026/Conference — ICLR 2026 Conference Withdrawn Submission_

### Official Review · Reviewer_MwWM · 2025-10-23

**Soundness:** 2
**Presentation:** 3
**Contribution:** 2
**Rating:** 2
**Confidence:** 4

**Summary:**

This paper verifies the effectiveness of model steering technology in alignment scenarios such as mathematical reasoning and code generation, and proves through experiments that model steering can achieve stronger alignment effects and lower costs in few-sample scenarios.

**Strengths:**

1. The overall writing is clear.
2. The experiments demonstrate several important conclusions, including the effectiveness of model steering in alignment scenarios.

**Weaknesses:**

1. The method itself lacks innovation: Representational steering has been extensively researched and applied in the fields of safety and factual alignment, with examples such as CAA and RepE. The method proposed in this paper is nearly identical to these approaches, simply migrating the context to alignment scenarios like mathematical reasoning.
2. The arguments in Lines 88-99 and the experimental results in Fig. 1 do not support the hypothesis. For example, Line 94 claims that observing a low angle indicates “mostly consistent in direction”, but this may not actually be the case. For example, the chosen-rejected distribution may be conical. It is recommended to visualize the chosen and rejected responses separately, rather than simply using distance and angle statistics.
3. Section 2.3 lacks specific experimental analysis for the neccesity of selecting steering directions.

**Questions:**

Line 189 says that only the response part is considered, not the instruction, but the selected token in line 194 is post-instruction tokens. What is the input here? Is the response input into the template as an instruction? This is very strange.

---

### Official Review · Reviewer_iyhq · 2025-10-29

**Soundness:** 2
**Presentation:** 3
**Contribution:** 3
**Rating:** 4
**Confidence:** 4

**Summary:**

This paper proposes a training-free preference alignment method, PALRS (Preference Alignment via Residual Steering). PALRS exploits preference signals encoded in a model’s residual stream activations. Using only on-the-order-of-hundreds preference pairs, it computes difference-in-means vectors between chosen and rejected responses, and applies the resulting steering vectors via activation addition at inference time.

The authors show that PALRS outperforms both the baseline and DPO models on mathematical reasoning and code generation, while scarcely degrading performance on general “guardrail” tasks (ARC-C, HellaSwag, MMLU, TruthfulQA, WinoGrande). The study indicates that linear intervention in the residual stream offers an efficient, flexible, and training-free paradigm for preference alignment.

**Strengths:**

1.  PALRS is training-free, avoiding costly, time-consuming gradient-based optimization. Alignment is implemented by simple vector addition at inference, yielding low computational overhead that scales favorably with model size. The plug-and-play nature enables easy switching or composition of steering vectors without maintaining multiple full checkpoints.
2. Clear experimental design, includes target tasks, guardrail evaluations, DPO comparisons, and α-sensitivity analyses.
3. Well-articulated contribution with thorough details. Establishes a framework for residual-based alignment and provides a public reproduction repository. The supplementary materials document layer ranges, token positions, steering coefficients, and other settings across models in substantial detail.

**Weaknesses:**

1.  **Fairness of the DPO comparison**

This is the primary concern. The paper trains both PALRS and DPO with 100 preference pairs. However, DPO is a gradient-based method whose performance strongly depends on data scale. Training DPO with only 100 pairs likely leads to severe underfitting, which is not a typical or recommended regime for DPO. Thus, the conclusion “PALRS outperforms 100-sample DPO,” while impressive under extreme data scarcity, may mislead readers into thinking PALRS generally surpasses DPO.
A fairer, fuller comparison should include DPO at typical data sizes (e.g., 1k, 5k, 10k pairs) to show the trade-offs across data regimes.

2. **Missing baselines**

The main experiments compare only to the original model, and the additional experiments compare only to DPO. Important baseline classes are absent: (1) other lightweight alignment or preference-optimization variants (e.g., PPO); (2) more recent preference-optimization methods; (3) alternative steering approaches; (4) other training-free techniques such as decoding-time methods.

3. **Mismatch between motivation and evaluations**

The motivation is human preference alignment, yet the evaluations focus on coding and mathematics, which have objective correctness. Core preference areas that are more subjective or multi-faceted (e.g., helpfulness, visual faithfulness, ethical considerations) are not tested, so it remains unclear whether PALRS is effective on genuinely subjective preference alignment.

4. **Heuristic dependence of key hyperparameters; lack of theory and adaptivity**

The effectiveness of method heavily depends on 3 hyperparameters: the layer l, the token position i, and the steering coefficient alpha. These are chosen by grid search on the test set, which still incurs cost and lacks theoretical guidance. Besides, sensitivity is high, fig. 3 shows that increasing alpha from 0.8 to 1.0 can sharply degrade performance, sometimes below baseline. This oversteering risk hinders direct deployment without careful manual tuning.

5. **Insufficient ablations**

Layer ranges and token positions are core design choices, yet there are no ablations isolating their impact on steering effectiveness, nor analysis connecting the steering coefficient with model/task characteristics.

**Questions:**

1. Do you consider the comparison setting—training DPO with only 100 preference pairs—fair to DPO? If, in the rebuttal, DPO is trained with more data (e.g., 1,000 pairs) while PALRS still extracts its steering vector from 100 pairs, how would the two methods compare?

2. The paper lacks a schematic that summarizes the full PALRS pipeline. Why did you choose not to include one? Adding a figure that clearly illustrates the flow—from steering-vector extraction to inference-time activation addition—would substantially improve clarity.

3. Fig. 3 indicates sharp drops when alpha is large. What is the mechanism behind this oversteering phenomenon? The figure also shows that alpha is highly sensitive and its optimum varies by model and task. In practice, if users do not have a labeled validation set for grid search, how do you recommend selecting the alpha for a new task? Is there a more robust or adaptive strategy for setting alpha?

4. Why does Llama-3B exhibit relatively common degradation on guardrail tasks? Is this a model-specific artifact or a limitation of PALRS? How can this be mitigated?

---

### Official Review · Reviewer_Nzmg · 2025-10-31

**Soundness:** 3
**Presentation:** 3
**Contribution:** 2
**Rating:** 2
**Confidence:** 4

**Summary:**

The paper introduces PALRS: a training-free algorithm for preference fine-tuning with time-saving advantages. PALRS is posed as an effective, efficient, and data-efficient strategy in comparison to complex RLHF and DPO pipelines. This "plug-and-play" method is demonstrably faster than DPO on GSM8K and HumanEval evaluations. The key approach is taking a small dataset of preferred vs non-preferred completions, finding the average internal activation for these answers at a specific layer. Then using this difference as a "steering" vector component. At inference time, this vector is added to the model's residual stream.

**Strengths:**

- The paper's biggest strength is the efficiency derived from its training-free strategy. It only needs one forward pass over some 100 samples to create the steering vector, which is where its time-speedups can be attributed to. Other discussed methods require a full fine-tuning run to align.

- The flexible plug-and-play approach is attractive, since it poses a possibility to be applied in many general preference learning domains. The method is not restricted to only mathematical reasoning, coding, or summarization domains. Since it is just a vector, and not a "permanent change", you can have a "math steering vector" and a separate "code steering vector", etc. This possibility for generalization is a strength.

- Since it only requires very few data point pairs (on the order of 100 samples) to create the vector, this becomes very data-efficient. This is a welcome angle from most RLHF and DPO regimes where "data is king" and tens of thousands of data point pairs are needed to fine-tune. It is aligned with the community's current interest in low data resource scenarios, such as medical settings.

**Weaknesses:**

- I have some concerns over the efficacy of the method. The simple vector-steering strategy does not permanently change the weights of the model, how effective can a training-free setting be? The performance gain comparisons with DPO only using 100 samples is not surprising, and ultimately even the most data efficient strategy is not relevant if it aligns poorly.

- Generalizability to larger models is unexplored. It is unclear if a simple "linear push" can steer a larger model towards the preferred outcome. The method for selecting the "best layer" to build the vector is also heuristics driven. It is unclear if these results can be reproduced.

- Current experiments focus on activation addition, and not directional ablation. Therefore this strategy becomes limited, encouraging certain outcomes, but cannot remove outcomes ((like harmfulness or toxicity). It is unclear if this is sufficient to align output in domains where completions are not boolean "right" versus "wrong". Especially since human preference tasks have become increasingly more complex in recent applications [1].


[1] Winata, Genta Indra, et al. "Preference tuning with human feedback on language, speech, and vision tasks: A survey." Journal of Artificial Intelligence Research 82 (2025): 2595-2661.

**Questions:**

- How do you determine finding the "best layer" in each model, and have the authors performed more principled analysis to define the "best layer" for building the steering vector?

- What does the rescaling in Eq 4 actually accomplish? This is rescaled to match the norm of the average chosen activation, and the claim is that this makes it comparable. But since this is immediately then multiplied by the  coefficient, the rescaling seems redundant. It is also unclear why Eq 3 aims to find a difference vector that is most aligned with average chosen activation. Is there any justification for this target? Why not just pick the vector with the largest magnitude, or even the most non-aligned rejected activations?

- The above points, as well as the introduction of hyperparameters  seem largely heuristics driven. Can the authors give some interpretability to these choices? How might the results be replicated, when complete grid-search for every introduced hyperparameter seems infeasible.

---

### Note · Authors · 2025-12-01

**Comment:**

We greatly appreciate the reviewers' comments and the mostly constructive suggestions they provided. Some of these, in particular, are important and non-trivial to address—or finalize, in the case of those we are already working on—within this rebuttal, and therefore merit careful consideration and development to ultimately help us improve and refine our work for future submission.

**Withdrawal Confirmation:**

I have read and agree with the venue's withdrawal policy on behalf of myself and my co-authors.